# Mucus Trail Proteins May Infer Reproductive Readiness for Land Snails

**DOI:** 10.3390/biology14030294

**Published:** 2025-03-14

**Authors:** Kate R. Ballard, Tomer Ventura, Tianfang Wang, Abigail Elizur, Scott F. Cummins

**Affiliations:** 1Centre for Bioinnovation, University of the Sunshine Coast, Sippy Downs, QLD 4556, Australia; kballard@usc.edu.au (K.R.B.); tventura@usc.edu.au (T.V.); twang@usc.edu.au (T.W.); aelizur@usc.edu.au (A.E.); 2School of Science, Technology and Engineering, University of the Sunshine Coast, Sippy Downs, QLD 4556, Australia

**Keywords:** pest, invasive land snail, gene expression, RNA-seq, reproduction

## Abstract

Land snail mucus trails have long been presumed to harbour semiochemical markers that influence trailing snail behaviour. Yet, there had been no experimental evidence found on what those markers may constitute. In this study, we analyzed the trail mucus protein profile of the land snail *Cernuella virgata*, comparing the reproductive stages and demonstrating distinct differences. As the result of the most significance, we report the inclusion of numerous albumen gland-derived proteins in trail mucus that are specific to reproductive-stage snails. Since the albumen gland is a key gland involved in reproduction, and specifically in the preparation of fertilized eggs prior to laying, we speculate that their inclusion into trail mucus may influence conspecific behaviour through the recognition of reproductive readiness. This finding has potential implications for the development of natural biocontrol agents for invasive land snails.

## 1. Introduction

The vineyard snail *Cernuella virgata* is native to the Mediterranean area of Europe, yet has become an invasive agricultural pest in many countries, including Australia, Israel, South Africa, and Spain [1]. In Australia, since their introduction in the 1900s and in concert with other land snails, they have become the eighth most economically damaging invertebrate agricultural pest, costing in excess of AUD 30 million annually in crop losses [2], with additional expenses relating to control that are difficult to measure. While various methods of control have been developed and undertaken, such as baiting, along with mechanical methods (e.g., rolling and slashing), they continue to be a significant economic burden on the grain industry both in Australia and worldwide.

In Australia, *C. virgata* breeds from autumn and early winter to spring, with most egg clutches laid in the first few months. As simultaneous hermaphrodites, these snails have an impressive breeding capacity, with the ability to lay thousands of eggs per season [3,4,5]. With the onset of late spring or early summer (November–December), the snails escape the increasing heat by climbing posts, grasses, weeds, or crop stalks. This coincides with the harvest period, leading to contamination of the grain by snails and their mucus [1,6,7]. The snails emerge from their aestivation during autumn (May), ready to feed, mate and lay eggs [7]. This period between autumn/winter and late spring is their reproductive stage, and more understanding of how this period differs from the non-reproductive stage at the molecular level may offer alternate approaches to mitigating current and future invasions. For example, a molecular analysis of components that drive olfactory-driven behaviours, such as pheromones, may provide new understanding that could be used to disrupt mate-finding. Pheromone attractant traps have proven successful as an approach for insect pest biocontrol [8,9]; however, due to our limited knowledge of pheromones in land snails, this approach is currently not an option.

A unique feature of gastropods is their ability to secrete mucus that can be used to facilitate locomotion, homing, and mate-finding [10,11]. For example, the marine gastropod, *Aplysia californica*, secretes a mucus during egg-laying that contains a cocktail of proteinaceous pheromones to attract conspecifics for mating purposes [12,13,14,15]. Those attractant proteins are derived from the animal’s albumen gland, a gland present in all pulmonate snails (including land snails), that surrounds fertilized eggs with perivitelline fluid [16]. A number of studies on land snails support the idea that mucus trails can be detected by conspecifics and convey information that leads to trail-following [17,18,19,20]. For example, in an investigation of trail-following in Hawaiian tree snails, five different species were shown to follow the trails of conspecifics, with *Partulina variabilis* following trails of conspecifics 94% of the time [20]. While trail-following has not been observed in *C. virgata*, recent work has shown that the activity and survival of *C. virgata* is reduced in the presence of mucus from another land snail, *Theba pisana*. This suggested that components of heterospecific mucus trail are detected and responded to by *C. virgata* [21].

Despite the recognition that trail mucus may contain components that facilitate trail-following and is potentially relevant to finding conspecifics for mating, the identity of detectable molecules in trail mucus remains largely unknown. Most investigations of snail mucus have been primarily focused on components’ medical utility and adhesive properties and have not examined the trail mucus specifically for snail-associated bioactivity [22,23,24,25]. Evidence suggests that snail mucus composition differs according to its function [26], leaving trail mucus a relatively under-researched commodity. A recent investigation of the common garden snail, *Cornu aspersum*, found that its trail mucus was made up of a large diversity of different proteins, including structural, enzymatic, and those that were considered novel [27], suggesting that it may not just be a medium to facilitate locomotion. In support of this, conspecific trail mucus did stimulate a significant increase in heartbeat rate upon contact. Similarly, structural, novel, and uncharacterized proteins have been identified in *C. aspersum* locomotive mucus [28]. However, neither study considered the snail’s reproductive stage, source of secretion, or interspecies variation.

This study performed a proteomics analysis of the *C. virgata* trail mucus at reproductive and non-reproductive stages, then integrated the findings with comparative transcriptomic analysis. The results further support the complexity of land snail trail mucus and variation that exists between the reproductive and non-reproductive stages. This knowledge could assist in the development of alternate, sustainable, and effective biocontrol strategies for *C. virgata* and other pest land snails.

## 2. Materials and Methods

### 2.1. Animals

Adult *C. virgata* were collected from three locations on the Yorke Peninsula (South Australia) during May and November 2019 and 2020 (co-ordinates: −34.75, 137.56; −34.91, 137.05; −34.64, 137.61). Snails were transported to the University of the Sunshine Coast, Sippy Downs (Queensland), where they were kept in aerated boxes, fed with carrot and lettuce, and sprayed daily with water.

### 2.2. Reference De Novo Transcriptome Assembly and Protein Annotation

A single *C. virgata* collected during May 2019 was euthanized and its shell was removed (total weight 0.8 g, shell diameter 12 mm). The snail’s flesh was finely sliced with a clean scalpel blade and transferred into 1.5 mL microtubes. After weighing, Trizol reagent (ThermoFisher Scientific, Scoresby, Australia) was added and total RNA was isolated according to the manufacturer’s instructions. The total RNA yield was determined using a Nanodrop spectrophotometer 2000c (Thermo Scientific, Waltham, MA, USA) at 260 and 280 nm. RNA was sent to Novogene (Beijing, China) for quality control and Illumina NovaSeq paired-end sequencing (150 bp) at 12 Gb. Clean data (clean reads) were screened from raw sequencing reads based on the following: (1) discard reads with adaptor contamination; (2) discard reads when uncertain nucleotides constituted more than 10% of either read (*n* > 10%); and (3) discard reads when low-quality nucleotides (base quality less than 20) constituted more than 50% of the read. Sequence datasets were deposited in the National Center for Biotechnology Information (NCBI) Bioproject (PRJ) database under accession number PRJNA858108.

High-quality reads were de novo assembled using SOAPdevono2 (CLC genomics workbench, version 10.1, Qiagen, Hilden, Germany) with parameters set as follows: seqType, fq; minimum kmer coverage = 4; minimum contig length of 100 bp; group pair distance = 250. Estimation of transcript expression was performed using the de novo RNA-Seq analysis tool on the CLC Genomic workbench software (Qiagen, version 23.0) with default parameters. A transcriptome-derived protein database was prepared using the open reading frame (ORF) finder (https://www.ncbi.nlm.nih.gov/orffinder/). In addition, this database was utilized to construct a protein secretome database using SignalP 5.0 (https://services.healthtech.dtu.dk/services/SignalP-5.0/) and transmembrane helices hidden Markov model (TMHMM, version 2.0), downloaded from http://www.enzim.hu/hmmtop/html/download.html.

### 2.3. Trail Mucus Collection and Protein Preparation

*Cernuella virgata* from June (reproductive stage) and November (non-reproductive stage) were washed in tap water and allowed to crawl in a 12 cm glass Petri dish for 1 h. Four dishes were used for each collection, with a maximum of 8 snails per dish. Snails were returned to the centre of the dish with forceps when they reached the perimeter and replaced with fresh snails when they were inactive for 5 min, to ensure that mucus was collected from approximately 30 individual snails. After removing any visible excrement with forceps, mucus for each species was collected into a 15 mL tube with the aid of a razor blade and a small amount of Milli-Q water and lyophilised overnight in a Speedvac Concentrator (SC250EXP; ThermoFisher Scientific, Scoresby, Australia). Lyophilised mucus was resuspended in 100 μL of Milli-Q water. Tubes were centrifuged for 5 min at 16,900× *g*, then, for each sample, the pellet and supernatant were placed into separate tubes, then quantified using a Nanodrop spectrophotometer at A280. Both pellet and supernatant proteins were size fractionated by 1D sodium dodecyl sulfate–polyacrylamide gel electrophoresis (SDS-PAGE) using 20 µL of sample (approx. 20 μg) in a Mini-Protean Tris-Glycine eXtended (TGX) (BioRad, Gladesville, Australia) gel, followed by Coomassie Blue staining. Protein sizes were estimated using a Pierce Blue Molecular Weight marker (ThermoFisher Scientific, Scoresby, Australia). The gel was imaged using a Bio-Rad Gel Doc XR^+^ Image lab (black and white; BioRad, Gladesville, Australia). Gel lanes were removed using a sterile scalpel and processed following a protocol described previously [29], in preparation for Liquid Chromatography Tandem Mass Spectrometry (LC-MS/MS).

### 2.4. LC-MS/MS and Protein Identification

LC-MS/MS was performed following the protocol described previously by Wang et al. [29]. Proteins detected using LC-MS/MS were matched to the whole *C. virgata* transcriptome database and all sequences were analyzed using NCBI Blastp to record the closest match. To determine which proteins were stage-specific, proteins identified in the trail mucus in both reproductive and non-reproductive stages were removed. Proteins unique to each stage were further filtered using SignalP 5.0 (https://services.healthtech.dtu.dk/service.php?SignalP-5.0) to retain secreted proteins, based on a probability of >0.5. Finally, genes for secreted proteins were checked for significant upregulation (*p* < 0.05) based on Fragments Per Kilobase per Million mapped fragments (FPKM) values in target tissues for the reproductive stage.

### 2.5. Reproduction Stage-Specific Tissue RNA-Seq Analysis

The albumen glands, mucous glands, and foot tissue of *C. virgata* were collected and carefully removed during May 2020 (reproductive) and November 2020 (non-reproductive), then immediately snap frozen on dry ice. Each biological replicate contained tissue from 5–10 individual snails. Total RNA was isolated with Trizol reagent according to the manufacturer’s instructions. Illumina sequencing was performed on 3 biological replicates for each reproductive stage, as described above to a depth of at least 10 Gb for each sample. Sequence datasets were deposited in the NCBI Sequence Read Archive (SRA) database under accession number PRJNA858108. Expression levels were calculated using RSEM (RNA-seq by Expect-Maximisation) and converted to FPKM, resulting in lists of genes upregulated and downregulated between reproductive stages. Differentially expressed genes were classified as significant if *p* < 0.05 with at least a ±2 log_2_ fold-change (FC). Significantly upregulated genes were further analyzed using the following: (1) gene ontology using 7 databases (NT, NR, Swissprot, KOG, PFAM, GO, and KEGG) with e-value thresholds from 0.01 to 1 × 10^−6^ and (2) tBLASTn against the in silico predicted secretome, to identify secreted proteins.

## 3. Results

### 3.1. Comparison of Reproductive- and Non-Reproductive-Stage Trail Mucus Proteins

The SDS-PAGE fractionation of *C. virgata* trail mucus pellet and supernatant extracts (Figure 1A), followed by mass spectrometry, led to the identification of 342 proteins at the reproductive stage and 307 proteins at the non-reproductive stage (Figure 1B and Appendix A). The majority of proteins were identified in trail mucus pellet preparations, and primarily consisted of structural proteins including collagen, tubulin, matrilin, actin, perlucin, tektin, and dynein. A total of 116 proteins were found in both stages of trail mucus. In the reproductive stage-specific trail mucus, 20 predicted secreted proteins were identified, including those with significant similarities to perlucin, achacin, and leukocyte elastase inhibitor (LEI). In the non-reproductive trail mucus, thirty-four stage-specific proteins were predicted to be secreted proteins, of which seven had no match on the NCBI database, while a further six matched to uncharacterized proteins in *A. californica* or *Biomphalaria glabrata*. The remainder of the unique non-reproductive trail mucus proteins detected in trail mucus included structural proteins such as matrilin and keratin, along with enzymes such as zinc metalloproteinase, endoglucanase, and nicotinamidase.

### 3.2. Overview of Tissue RNA-Seq Data and Differential Gene Expression

To determine the source and temporal gene expression of trail mucus proteins, transcriptomic analysis of the *C. virgata* albumen gland, mucous gland, and foot (all secretory tissues) was investigated during the reproductive and non-reproductive stages. A principal component analysis of the RNA-seq data demonstrated a distinct segregation of reproductive stages, particularly for the albumen gland and mucous gland (Figure 2). Subsequent differential gene expression analysis showed a comparatively lower number of genes significantly upregulated during the reproductive stage within the gland tissues; however, many demonstrated a higher fold-change increase (Figure 3A and Appendix A). For example, of the upregulated albumen gland genes, 72 had >10 log_2_ fold-change, including 29 novel genes (with no match in the NCBI database) and 14 genes that encoded uncharacterized gastropod-like proteins. Gene ontology analysis demonstrated that, overall, the reproductive stage tissues expressed an enrichment of genes associated with transport (transmembrane transport and transmembrane transporter activity) and enzymatic activity (transferase, phosphatase, peptidase, oxidoreductase, and ligase) (Figure 3B).

### 3.3. Spatial and Temporal Gene Expression of Identified Secreted Proteins in Trail Mucus

Identified secreted proteins that were exclusive to the non-reproductive stage trail mucus (Figure 1B) demonstrated a gene expression pattern that was largely specific to the non-reproductive stages of the albumen and mucous glands (Figure 4A). Meanwhile, those expressed in the foot were equally expressed in both reproductive stages. The genes with the highest level of expression at the non-reproductive stage were matrilin, a collagen altha-1, a protein NPC2 homologue, a bactericidal permeability-increasing protein, and various novel genes. Identified trail mucus-secreted proteins that were exclusive to reproductive stage (Figure 1B) were significantly upregulated in only the albumen gland, and with minimal to no gene expression in the non-reproductive stage albumen gland, as well as other tissues analyzed, irrespective of reproductive stage (Figure 4B). Those genes with the highest level of expression in the reproductive stage included the leukocyte elastase inhibitors (LEI), achacin, perlucin, and several uncharacterized genes.

## 4. Discussion

Despite the global pervasiveness of invasive snails and the economic damage they inflict, little is known about snail communication, and whether the trail mucus might play a role. This study used proteo-transcriptomic analysis of the pest land snail *C. virgata*, combining trail mucus with differential gene tissue expression to determine whether any trail mucus proteins could potentially function as pheromones. The results indicated large differences in the mucus trails between reproductive stages. As semiochemicals would necessarily be secreted from the cell, the focus was on secreted proteins. We identified proteins that may be involved in intra- and inter-species communication, which appear to be largely derived from the albumen gland during the reproductive season.

### 4.1. Proteins Detected in Non-Reproductive-Stage Trail Mucus

Of the predicted secreted proteins that were unique to the *C. virgata* non-reproductive trail mucus, seven had no significant database matches. The remainder comprised six uncharacterized proteins, several structural proteins such as matrilin, collagen, and keratin, and a number of enzymes, including zinc metalloproteinase, nicotinamidase, and endoglucanase. Matrilin was also identified in the albumen gland and egg mass of the freshwater snail, *B. glabrata*, where it is thought to act as a putative defence molecule [30]. Zinc metalloproteinase was also found in the snail-conditioned water of *B. glabrata*, where it is believed to reduce tissue damage and inflammation [31]. Also present were sialic acid binding lectin, an immunoglobulin-like lectin that promotes cell death, along with a leucine-rich repeat protein and an NPC2 homologue from *B. glabrata.* Indeed, it is conceivable that most, if not all, of the abovementioned proteins function in an immunity-related capacity in the mucus trail, where they function in the protection and repair of the foot. Four of the seven novel proteins in the non-reproductive trail mucus were only expressed in the albumen gland during the non-reproductive stage (contigs 10553, 12028, 14919, and 29098).

In this study, similarities were found with recent research into *C. aspersum* mucus, particularly in structural and gel-forming proteins such as mucin, collagen, fibrinogen, and epiphragmin [27,28]. Along with the current study, these analyses included a large number of uncharacterized and novel proteins, termed snail-related proteins, that require further work with a view to elucidating their functions. However, it should be noted that some of these proteins may be species-specific, and neither of the *C. aspersum* mucus studies were season-specific.

### 4.2. Proteins Detected in the Reproductive-Stage Trail Mucus

The analysis of the mucus trail of *C. virgata* yielded 226 proteins that were only present during the reproductive stage. These included predicted secreted proteins, of which 12 matched to uncharacterized proteins. Of the remaining proteins, one interesting finding was the identification of perlucin, a shell matrix protein identified in a number of mollusc species, including abalone, oyster, mussel, and snail species [27,32,33,34,35]. However, other studies suggest that the function of perlucin might be more diverse than previously thought. Whaite et al. identified perlucin in the byssal thread of the pearl oyster *P. maxima*, suggesting that it may be important in adhesion between organic surfaces [35]. Some studies suggest that the C-lectin binding domain in this protein may function in linking the living tissue to biomaterial, such as adhesive mucus or byssal thread [35,36]. Alternatively, as a C-type lectin, this protein may play a role in immunity [37], which is an important function of mucus, as it interfaces with the surrounding environment. In addition, perlucin was recently detected in the trail mucus of the common garden snail, *C. aspersum* [27,38], suggesting a specific role in the mucus trail, although reproductive status was not assessed. Cerullo et al. [28] also described a large number of lectin-type proteins, in both the lubricating and protective mucus of *C. aspersum*, including CD109 lectins. In the current study, the perlucin protein was highly expressed in the albumen gland, with a 9.3-fold increase in expression during the reproductive period. This could be explained by the necessity of this protein for egg formation. Once again, the presence of perlucin in the mucus trail could be an indicator of reproductive health, and the ability to lay viable eggs.

Achacin was another protein that was present only in the reproductive trail mucus, which was detected as highly upregulated in the albumen gland during the reproductive stage with a fold change of 11.3. Achacin is an L-amine oxidase, which was originally characterized in the mucus of the giant African land snail, *Lissachatina fulica* [37,39]. Its antimicrobial activity was thought to be due to its role in the catalysis of oxidative deamination of L-amino acids, which produces hydrogen peroxide as a by-product [39]. Achacin homologs have been detected in the mucus of other molluscs, such as the sea hare, *Aplysia punctata*, in which the protein is called ‘cyplasin’ [40]. Due to its relative abundance in reproductive-stage land snails, it may be an important component of the eggs, functioning to protect the eggs both pre- and post-laying. As *C. virgata* lay their eggs in soil, protection from microbes is an important factor in hatching success. Consistent with this theory, an achacin homologue has been detected in egg cordons of the aquatic snail *B. glabrata* [30]. Similarly, the related aplysianin-A is produced in the albumen gland of the sea hare *Aplysia kurodai* and is found on the egg cordons once deposited on the sea floor [41]. Due to its crucial health-related functions, the detection of this protein in the mucus may provide important information to conspecifics on the general health of the trail-laying snail, and the probability of egg survival and hatching.

The reproductive-stage albumen gland also revealed the high upregulation of trail mucus LEI, compared to the non-reproductive-stage albumen gland. While it has been found to function as an antibacterial defence molecule, it also plays a role in maintaining sperm motility [42] and is expressed in the human reproductive system, including the female fallopian tubes and male semen [42]. It was also detected in the water around aquatic snails, *B. glabrata*, that were resistant to *Schistosoma mansoni* infection, further suggesting an immune defence function [31]. A similar protease inhibitor has been reported from the albumen gland and fractions of egg masses of the pond snail *Lymnaea stagnalis*, suggesting that this protein plays an important role in reproduction, and may be crucial to the protection of the embryo during development [43]. The presence in the reproductive tract along with its putative immune function and high levels in the albumen gland during the reproductive stage are strongly suggestive of a role in egg protection, at the very least. Similarly to perlucin and achacin, detection in the mucus trail by conspecifics could attest to the reproductive health and viability of fertilised eggs of the prospective sexual partner. Serine protease inhibitors were also identified in the locomotive mucus of *C. aspersum* [28].

Another notable protein exclusive to reproductive-stage mucus trail was encoded by the gene contig 8667, a protein with unknown function that was most abundantly expressed in the albumen gland. The highest database match for this sequence related to a hypothetical protein in *Elysia chlorotica* (3 × 10^−32^); however, at only 26% identity, this identification is questionable. Other matches, such as capsulin from *Elysia marginata*, seem more likely, although the identity percentage is still only around 25%, suggesting that this protein is as yet an undescribed protein with a unique function. However, this protein contains numerous cysteine residues and the ‘EECK’ amino acid sequence, which is a distinct region of the *A. californica* attractin, a recognised pheromone that attracts conspecifics for mating. This raises the possibility that it may have a pheromonal function in this species. Calcium-binding proteins that were found to be abundant in the locomotive mucus of *C. aspersum*, such as calmodulin and EF-hand proteins [28], were also found in the mucus trail of *C. virgata*, with calmodulin only present in the reproductive stage mucus.

Our findings indicated that the albumen gland is only actively secreting reproductive-associated proteins into the mucus trail during the reproductive stage. While the purpose of this is unclear, it remains a distinct possibility that *C. virgata* communicates its reproductive status through its mucus trail, using proteins secreted by the albumen gland in order to attract a mating partner. This is consistent with the protein pheromones of the marine gastropod *A. californica*, which are produced from the albumen gland and released with the egg cordons [12,13,14,15,29]. There may be a similar mechanism in place with terrestrial snails, using the mucus trail as a semi-aqueous medium for protein deposition. In support of this, preliminary behavioural assays conducted using albumen gland extract demonstrated increased interest in filter paper soaked in the extract, along with increased activity. However, as these results were not measurable, they serve as a starting point for further behavioural investigations into the effect of this gland components on conspecific snails.

### 4.3. Genes Upregulated in the Albumen Gland at Reproductive Stage, Yet Not Identified in Trail Mucus

Aside from those proteins identified in the reproductive stage trail mucus, several genes that encoded predicted secreted proteins were found to be highly upregulated in the albumen gland of *C. virgata* during the reproductive stage, including agglutinin, cathepsin, and temptin. Both agglutinin and cathepsin can be linked with a role in defence. A member of the H-lectin superfamily, agglutinin was named for its ability to aggregate human red blood cells, along with certain bacteria and viruses. In land snails, agglutinin has been found in the albumen gland of the Roman snail, *Helix pomatia*, in which it is an important component of perivitelline fluid and is thought to protect fertilised eggs from bacterial infection [44]. A link between lectin activity and egg protection in the freshwater snail *Pomacea scalaris* was postulated, whereby it may also play a role in protection from predators post-laying [45]. Agglutinin-like proteins were detected in the mucus of *C. aspersum* [27].

Various members of the cathepsin family were upregulated in the albumen gland, primarily cathepsins B and L. Both of these cathepsins are cysteine proteases, which are involved in apoptosis and innate immunity, where they assist in the regulation of inflammation [46]. Cathepsin family members have been found in other mollusc species, including the Pacific abalone, *Haliotis discus hannai*, in which cathepsin B was found to be upregulated after bacterial infection [47]. The cysteine proteases were the most common cathepsins detected in the Mediterranean mussel *Mytilus galloprovincialis*, with an isoform of cathepsin L most abundant in the female gonad [48].

Snails in this study also showed significant upregulation of temptin, which encodes a protein that exhibits significant similarity to that of *A. californica* temptin. In *A. californica*, temptin is secreted from the albumen gland onto the egg cordon, where it is released with other protein pheromones into the surrounding seawater to attract conspecifics [15]. Temptin homologs have also been described in other molluscs, including the freshwater snail *B. glabrata* [49], where it was also found to attract conspecifics. However, in other mollusc species, it has been linked with shell biomineralization [50]. This could provide an alternative explanation for its upregulation in the albumen gland, where it could be utilised by the developing embryo to produce a shell prior to hatching.

## 5. Conclusions

This study has provided new information regarding the protein components present within the mucus trail of *C. virgata*, as well as reproductive stage-specific differences. The results have highlighted the possibility that, during the reproductive season, land snails secrete proteins from their reproductive systems into their mucus trails. A large number of albumen gland-derived mucus trail proteins have immunity and defence functions for developing embryos within eggs. Yet, when deposited into mucus trails, these proteins may play a role in communicating a snail’s reproductive status, subsequently enhancing reproductive success. Future research should explore whether one or more of these proteins could be utilised to modify snail behaviour (e.g., attractant), which could lead to a novel biocontrol strategy.

## Figures and Tables

**Figure 1 biology-14-00294-f001:**
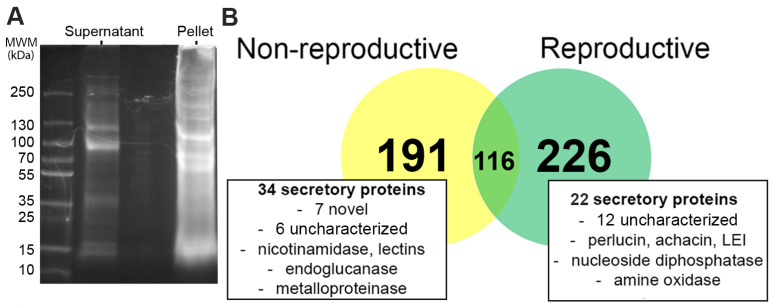
Comparative analysis of *Cernuella virgata* trail mucus proteins at non-reproductive and reproductive stage. (**A**) SDS-PAGE with Coomassie stain gel showing separation of isolated trail mucus proteins from pellet and supernatant (See Appendix A). (**B**) Venn diagram showing shared and unique proteins found in trail mucus. See Appendix A for annotation of trail mucus proteins identified in reproductive and non-reproductive stage *C. virgata*. See Appendix A for protein sequences identified in the reproductive and non-reproductive trail mucus, respectively.

**Figure 2 biology-14-00294-f002:**
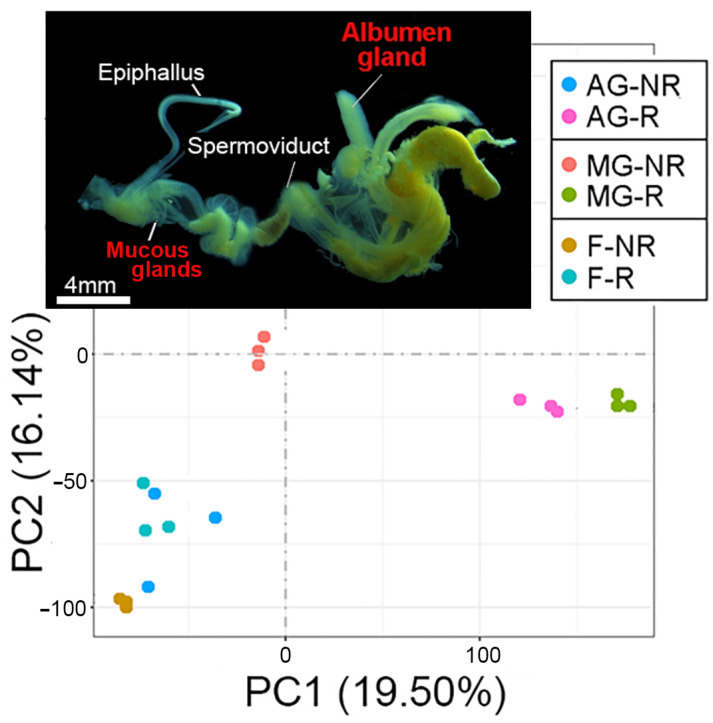
Principal component analysis (PCA) of tissue-specific RNA-seq for *Cernuella virgata*. Image shows the dissected reproductive system of *C. virgata* (including target albumen gland and mucous glands). PCA demonstrates the distribution of RNA-seq data for the albumen gland, mucous gland, and foot tissue during the non-reproductive (NR) and reproductive (R) stages.

**Figure 3 biology-14-00294-f003:**
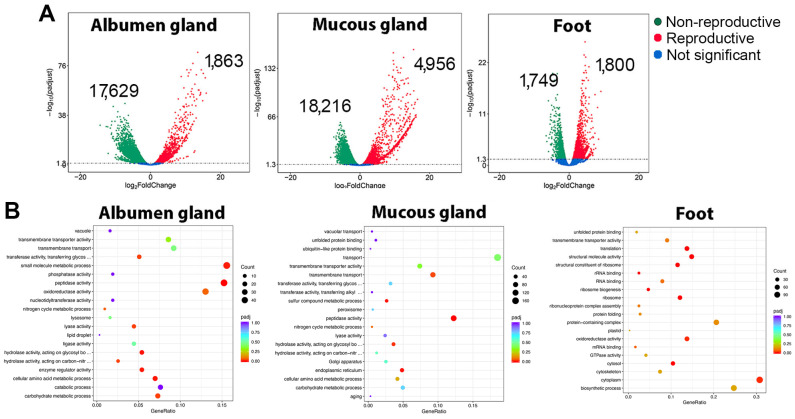
Differential gene expression and functional analysis in *Cernuella virgata* tissues. (**A**) Volcano plots showing the number of significantly upregulated genes during non-reproductive and reproductive stage. See Appendix A for the total list of significantly (FDR *p* < 0.05) differentially expressed genes. (**B**) Top 20 gene ontology results for significantly up-regulated genes during the reproductive stage of each tissue.

**Figure 4 biology-14-00294-f004:**
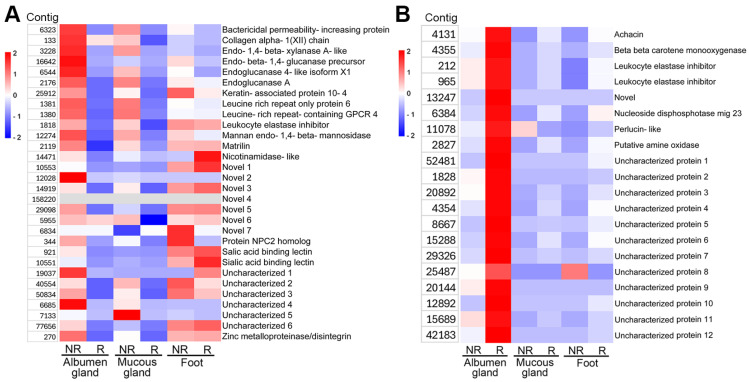
Assessment of tissue gene expression for *Cernuella virgata* trail mucus proteins. Heatmaps showing the comparative tissue gene expression (FPKM) for predicted secreted proteins found in the (**A**) non-reproductive stage (NR) trail mucus, and (**B**) reproductive (R) stage trail mucus.

## Data Availability

The RNA-seq assemblies are available from PRJNA850094 and PRJNA858108.

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
