# Peer review of "Mucus Trail Proteins May Infer Reproductive Readiness for Land Snails"

_biology, 2025, doi:10.3390/biology14030294_

Round 1
Reviewer 1 Report
Comments and Suggestions for Authors
This manuscript by Ballard K.R. and colleagues reveals the protein and gene expression profiles of the land snail C. virgata, with an emphasis on its reproductive state. Their proteomic and transcriptomic analyses identified a variety of secretory proteins expressed in the albumen gland, one of the reproductive organs, during the reproductive phase.
Overall, the manuscript is well-written and easy to follow. However, several points should be considered by the authors to fully support their conclusion.
[1] Introduction: The rationale for focusing on the reproductive state in C. virgata is lacking.
I feel there isn't enough rationale, supported by behavioral observations, to compare snail mucus in either a reproductive or non-reproductive state. The specific species used by the authors in their study, C. virgata, is apparently not an ideal model since there is no evidence that this snail exhibits trail-following behavior (as mentioned, "While trail following has not been observed in C. virgata, ..."), and the potential ability to sense the presence of other snail species does not guarantee that C. virgata itself produces special mucus in the reproductive state. If the authors wish to connect their findings to something specific to reproduction, they should provide evidence of reproduction-related behaviors (e.g., trail-following) in C. virgata. Otherwise, they should explain why they chose to use C. virgata instead of other snail species with documented evidence of trail-following, and detail how they could apply this study to those species.
[2] Results: Gene expression information in reproductive organs other than the albumen gland is required to support their notion.
The authors focused on the albumen gland primarily due to the clear separation between reproductive vs non-reproductive clusters in the PCA analysis (Fig 2A). However, it remains unclear whether the changes in gene expression within the albumen gland significantly contribute to the protein enrichment found in the mucus derived from the reproductive state, as analyzed in Fig 1. To support their conclusion, "The results have highlighted the possibility that during the reproductive season, land snails secrete proteins from the reproductive system into the mucus trail," the authors should perform comparative analyses between the albumen gland and other organs (such as the mucous gland and foot). This can be achieved by using their transcriptome data or by adding independent qRT-PCR tests targeting the six genes commonly detected in both analyses (marked with asterisks in Fig 2D). The data should be clearly presented in the main figure, demonstrating that the six genes marked with asterisks in Fig 2D are most abundantly expressed in the albumen gland, and the expression changes in this organ should largely account for the protein enrichment in the secreted mucus.
[3] Results: Analyses of the proteins/genes downregulated in the reproductive state may provide valuable insights.
The authors focus solely on the genes that are upregulated in the reproductive state, as shown in Fig 2D. However, there is a possibility that an "inhibitor" of trail following is present in the mucus during the non-reproductive state, and its downregulation during the reproductive state could relate to the snails' social behaviors. It would be informative if the authors could include a table similar to Fig 2D that displays the highly down-regulated genes and mark them with asterisks if they are also detected in the non-reproductive mucus from the protein analyses (Fig 1).
[4] Results: Protein sequence comparison should be informative for the six genes commonly identified in both analyses.
If the authors argue that the mucus protein identified in their study may play a role in inter-species communication, it makes sense to conduct protein sequence comparisons for the six genes shown in Fig 2D, which are commonly found in both protein and gene-expression analyses. One expectation is that these six genes encode proteins that are not conserved in related species, especially Theba pisana, since a report suggests that C. virgata is capable of distinguishing itself from this species (according to the authors' introduction).
On the other hand, the protein sequence comparison for temptin (Fig. 3) has relatively less importance and could therefore be omitted. The LC-MS/MS in Fig. 1 failed to detect temptin, resulting in a discrepancy between protein and gene expression analyses.
[5] Methods: Details for protein preparation are lacking.
Fig 1A indicates that "The majority of proteins were identified in trail mucus pellet preparations," but it remains unclear whether the authors utilized the pellet or the supernatant for the LC-MS/MS analyses. In methods section 2.3, it states, "Lyophilised mucus was resuspended in 100 ml of Milli-Q water. Tubes were centrifuged for 5 min at 16,900xg, then for each sample the pellet and supernatant were placed into separate tubes." However, they did not specify which sample (pellet or supernatant) was used in the subsequent steps. Please clarify these details in the methods section.
Author Response
[1] Introduction: The rationale for focusing on the reproductive state in C. virgata is lacking.
I feel there isn't enough rationale, supported by behavioral observations, to compare snail mucus in either a reproductive or non-reproductive state. The specific species used by the authors in their study, C. virgata, is apparently not an ideal model since there is no evidence that this snail exhibits trail-following behavior (as mentioned, "While trail following has not been observed in C. virgata, ..."), and the potential ability to sense the presence of other snail species does not guarantee that C. virgata itself produces special mucus in the reproductive state. If the authors wish to connect their findings to something specific to reproduction, they should provide evidence of reproduction-related behaviors (e.g., trail-following) in C. virgata. Otherwise, they should explain why they chose to use C. virgata instead of other snail species with documented evidence of trail-following, and detail how they could apply this study to those species.
RESPONSE:
We have added text that better rationalises the use of comparative non-reproductive and reproductive samples – to the abstract and introduction. Such as:
Abstract - This is particularly relevant during periods of reproduction, whereby conspecific cues are critical towards finding mating partners.
Introduction - Despite recognition that trail mucus may contain components that facilitate trail following, potentially relevant to finding conspecifics for mating, the identity of detectable molecules in trail mucus remains largely unknown.
In addition, we acknowledge that C. virginica does not have any recorded evidence for trail following, and therefore our purpose was to assess whether it may show differences in trail mucus protein profiles based on reproductive season. C. virgata was chosen due to: 1. The dearth of research for this species, 2. The significance of this species as a grain pest in Australia. 3. Its close relationship to two other significant grain pests. Ie Cochlicella acuta and Cochlicella barbara are same family – Hygromidae.
[2] Results: Gene expression information in reproductive organs other than the albumen gland is required to support their notion.
The authors focused on the albumen gland primarily due to the clear separation between reproductive vs non-reproductive clusters in the PCA analysis (Fig 2A). However, it remains unclear whether the changes in gene expression within the albumen gland significantly contribute to the protein enrichment found in the mucus derived from the reproductive state, as analyzed in Fig 1. To support their conclusion, "The results have highlighted the possibility that during the reproductive season, land snails secrete proteins from the reproductive system into the mucus trail," the authors should perform comparative analyses between the albumen gland and other organs (such as the mucous gland and foot). This can be achieved by using their transcriptome data or by adding independent qRT-PCR tests targeting the six genes commonly detected in both analyses (marked with asterisks in Fig 2D). The data should be clearly presented in the main figure, demonstrating that the six genes marked with asterisks in Fig 2D are most abundantly expressed in the albumen gland, and the expression changes in this organ should largely account for the protein enrichment in the secreted mucus.
RESPONSE:
Our initial goal was to promote as quicky as possible the albumen gland as the primary tissue that sourced the reproduction-derived trail mucus proteins. However, we acknowledge that there were aspects that should be unwrapped prior to this, to ensure clarity. As such, we have now included in the main text a PCA that includes RNA-seq for all tissues analysed, and at both reproductive stages (Figure 2), as well as their differential gene expression and enriched gene ontology (Figure 3). We thank the reviewer for recognising that this is of interest as part of the overall conclusions.
We believe that the proteomic evidence, supported by quantitative RNA-seq, provides demonstratable evidence of their importance (without qRT-PCR).
[3] Results: Analyses of the proteins/genes downregulated in the reproductive state may provide valuable insights.
The authors focus solely on the genes that are upregulated in the reproductive state, as shown in Fig 2D. However, there is a possibility that an "inhibitor" of trail following is present in the mucus during the non-reproductive state, and its downregulation during the reproductive state could relate to the snails' social behaviors. It would be informative if the authors could include a table similar to Fig 2D that displays the highly down-regulated genes and mark them with asterisks if they are also detected in the non-reproductive mucus from the protein analyses (Fig 1).
RESPONSE:
Again, this was our oversight to focus only on reproductive stage trail mucus, and relevant genes. In acknowledgement of that those downregulated are of interest, yet could relate to our trail mucus proteomic data, we have now included data that highlights genes/proteins that were observed during non-reproductive season. The outcomes have not changed for the study, yet do ensure the outcomes are of interest to a broader audience.
[4] Results: Protein sequence comparison should be informative for the six genes commonly identified in both analyses.
If the authors argue that the mucus protein identified in their study may play a role in inter-species communication, it makes sense to conduct protein sequence comparisons for the six genes shown in Fig 2D, which are commonly found in both protein and gene-expression analyses. One expectation is that these six genes encode proteins that are not conserved in related species, especially Theba pisana, since a report suggests that C. virgata is capable of distinguishing itself from this species (according to the authors' introduction).
On the other hand, the protein sequence comparison for temptin (Fig. 3) has relatively less importance and could therefore be omitted. The LC-MS/MS in Fig. 1 failed to detect temptin, resulting in a discrepancy between protein and gene expression analyses.
RESPONSE:
We have now removed Figure 2D and related results, which was biased towards the secretion of proteins from the albumen gland. To better align with a focus on the secreted proteins present in the mucus, both non-reproductive and reproductive trail mucus, we only address tissue differential gene expression, and finally those trail mucus proteins identified and tissue gene expression (Figure 4). We would like to address specific functional aspects of proteins in a separate study.
We have removed the Figure 3 Temptin. Although of interest in conspecific communication, it was not identified in the trail mucus. We keep discussion relevant to its presence as a differentially expressed gene.
[5] Methods: Details for protein preparation are lacking.
Fig 1A indicates that "The majority of proteins were identified in trail mucus pellet preparations," but it remains unclear whether the authors utilized the pellet or the supernatant for the LC-MS/MS analyses. In methods section 2.3, it states, "Lyophilised mucus was resuspended in 100 ml of Milli-Q water. Tubes were centrifuged for 5 min at 16,900xg, then for each sample the pellet and supernatant were placed into separate tubes." However, they did not specify which sample (pellet or supernatant) was used in the subsequent steps. Please clarify these details in the methods section.
RESPONSE:
We have updated the methodology to clarify the samples used, which better reflect the results shown in Figure 1A.
Tubes were centrifuged for 5 min at 16,900xg, then for each sample the pellet and supernatant were placed into separate tubes, then quantified using a Nanodrop spectrophotometer at A280. Both pellet and supernatant proteins were size fractionated by 1D sodium dodecyl sulfate-polyacrylamide gel electrophoresis (SDS-PAGE) using 20 µl of sample (approx. 20 mg) in a Mini-Protean TGX (BioRad) gel followed by Coomassie Blue staining.
Utlinmately, the final protein database was combined from both pellet and supernatant to obtain the maximum number of proteins that represent the trail mucus within non-reproductive and reproductive trail mucus.
Reviewer 2 Report
Comments and Suggestions for Authors
Dear Authors,
The manuscript entitled “Mucus trail proteins may infer reproductive readiness for land snails” is recommended by me for publication in the journal Biology.
The work entails identifying proteins in the albumen glands, mucous glands, and foot of the land snail Cernuella virgata during both reproductive and non-reproductive stages. A difference in protein composition was observed between these two stages, clearly indicating that certain proteins may play a role in reproductive readiness, reproductive health, the viability of fertilised eggs, or the production of reliever pheromones.
The C. virgata snail presents a significant biological and economic threat to agricultural fields in Australia and worldwide. This hermaphroditic species lays thousands of eggs during its breeding season, which occurs from autumn to early winter and into spring. Understanding the biology and biochemistry of this species is essential for implementing effective biological control methods in crops and gardens.
The work identified the upregulated activity of 13 genes encoding 13 uncharacterised proteins (Fig. 1B, 4, pp. 7); however, on page 6, it mentions 14 genes (?).
I am not a native speaker, but I suggest doing some English corrections:
1) In many places, it should be written “the reproductive stage” instead of reproductive stage, for example, in Abstract and Simple Summary, Fig. 4
2) Page 5 - Overview of RNA-seq data and differential gene expression in the albumen gland
To investigate the source and temporal gene expression of trail mucus proteins,
transcriptomic analysis of the C. virgata albumen gland, mucous gland and foot (all
secretory tissues) was investigated during the reproductive and non-reproductive stages.
Principal component analysis of the RNA-seq data demonstrated a distinct segregation of
reproductive stages (File S4), but most clearly in the albumen gland (Figure 2A).
3) Spelling “upregulated” - Fig.2 - Principal component analysis (PCA), differential gene expression and functional analysis of albumen gland-specific RNA-seq for C. virgata. (A) Image showing the dissected reproductive system of C. virgata (including target albumen gland) and PCA demonstrating the distribution of RNA-seq data for albumen gland. (B) Volcano plot showing up-and down-regulated genes in the albumen gland during the reproductive stage. See File S5 for the total list of significantly (FDR P<0.05) differentially expressed genes. (C) Top 20 gene ontology results for upregulated albumen genes. (D) Heatmap showing relative expression of genes with high e-value annotation support between albumen gland at reproductive (R) and non-reproductive (NR) stage. * denotes those proteins identified in trail mucus
4) Page 2, Paragraph 2 – it should be …., with most egg clutches are laid in the first few months.
5) This paper is written in American English, but in Figures 1 and 4 there is “uncharacterised” instead of uncharacterized.
6) Page 8, Paragraph 1 – please cut one “the” before the pest.
7) Page 10, Paragraph 1 – please correct the spelling of the word “pheromone”.
8) Page 10, Paragraph 3 – It should be ….the perivitelline fluid, and is thought to……
9) Page 11, Paragraph 1 - It should be …., where it is released with to……
10) Page 11, Paragraph 2 - It should be …. during the reproductive season, ……
11) In Simple Summary - It should be …. implications for the development of……
12) In Abstract - It should be …. key gland for the preparation of…….
The end of comments.
Author Response
The work identified the upregulated activity of 13 genes encoding 13 uncharacterised proteins (Fig. 1B, 4, pp. 7); however, on page 6, it mentions 14 genes (?).
RESPONSE:
After reassessment of the data, we now consistently report 12 uncharacterized proteins, reflected in Figure 1B and Figure 4B.
1) In many places, it should be written “the reproductive stage” instead of reproductive stage, for example, in Abstract and Simple Summary, Fig. 4
RESPONSE:
We have now amended to ‘the reproductive stage’ and ‘the non-reproductive stage’ to the text as needed.
2) Page 5 - To investigate the source and temporal gene expression of trail mucus proteins, transcriptomic analysis of the C. virgata albumen gland, mucous gland and foot (all secretory tissues) was investigated during the reproductive and non-reproductive stages.
RESPONSE:
We have updated the text accordingly.
3) Spelling “upregulated” - Fig.2 - Principal component analysis (PCA), differential gene expression and functional analysis of albumen gland-specific RNA-seq for C. virgata. (A) Image showing the dissected reproductive system of C. virgata (including target albumen gland) and PCA demonstrating the distribution of RNA-seq data for albumen gland. (B) Volcano plot showing up-and down-regulated genes in the albumen gland during the reproductive stage. See File S5 for the total list of significantly (FDR P<0.05) differentially expressed genes. (C) Top 20 gene ontology results for upregulated albumen genes. (D) Heatmap showing relative expression of genes with high e-value annotation support between albumen gland at reproductive (R) and non-reproductive (NR) stage. * denotes those proteins identified in trail mucus
RESPONSE:
We have updated the text accordingly, except for text that has been removed.
4) Page 2, Paragraph 2 – it should be …., with most egg clutches are laid in the first few months.
5) This paper is written in American English, but in Figures 1 and 4 there is “uncharacterised” instead of uncharacterized.
6) Page 8, Paragraph 1 – please cut one “the” before the pest.
7) Page 10, Paragraph 1 – please correct the spelling of the word “pheromone”.
8) Page 10, Paragraph 3 – It should be ….the perivitelline fluid, and is thought to……
9) Page 11, Paragraph 1 - It should be …., where it is released with to……
10) Page 11, Paragraph 2 - It should be …. during the reproductive season, ……
11) In Simple Summary - It should be …. implications for the development of……
12) In Abstract - It should be …. key gland for the preparation of…….
RESPONSE:
All suggestions have been corrected in the updated manuscript.
Round 2
Reviewer 1 Report
Comments and Suggestions for Authors
The authors have effectively addressed my concerns.
Author Response
Comment: The authors have effectively addressed my concerns.
Response: We thank the reviewer for recognising our changes have addressed all concerns.